# Understanding the Qubit Trajectory in Parameterized Quantum Circuits

Seungcheol Oh[1][0009−0007−8979−604X], Chaemoon Im[1][0009−0009−7270−7332], Soohyun Park[2][0000−0002−6556−9746], and Joongheon Kim[1][0000−0003−2126−768X]

[1] Korea University, Seoul 02841, Korea
{seungoh,anscodla0314,joongheon}@korea.ac.kr
[2] Sookmyung Women's University, Seoul 04310, Korea
soohyun.park@sookmyung.ac.kr

**Abstract.** Quantum neural networks (QNNs) employ parameterized quantum circuits (PQCs) to map data inputs to desired predictions. Similar to classical supervised learning, PQCs are often treated as black-box function mappers. Recent analytical research demonstrates that PQCs consisting of data reuploading structure are naturally expressed as partial Fourier series and that a single qubit circuit can serve as a universal approximator for univariate functions. However, this prior work largely focuses on representational capacity and does not provide intuitive or structural explanations of how the individual circuit components govern the resulting Fourier coefficients. In this paper, we peel back the black-box and analyze the intrinsic structure of PQCs. In particular, we investigate how data encoding, repeated reuploading, and trainable unitary operators combine to represent function classes characterized by a Fourier expansion with specific accessible frequency components. Our key contribution is to show, both mathematically and empirically, that the configuration of trainable parameters creates an output qubit trajectory that is critical for the representation of Fourier coefficients.

**Keywords:** Quantum Neural Networks · Parameterized Quantum Circuits.

## 1 Introduction

**Background and Motivation.** Quantum computing sits at the intersection between physics and computation. By leveraging the principles of quantum mechanics, such as superposition and entanglement, quantum computers can solve certain highly complex problems that are otherwise intractable to solve with classical computer, achieving what is known as a *quantum supremacy* [Deutsch and Jozsa(1992),Grover(1996),Shor(1999),Arute et al.(2019)]. Although the general notion of *quantum supremacy* evokes the ideas that quantum computers inherently outperform classical computers due to their quantum nature, such benefit does not arise automatically. Rather, it arises from the careful design of quantum algorithms that exploit specific quantum properties tailored to the

given problem. Thus, to fully realize the quantum advantage in any scientific domain, it is essential to deeply understand how quantum systems behave under the specific problem conditions so that it can be meaningfully incorporated into the problem's algorithmic designs [Li et al.(2022)]. Quantum computing is increasingly being applied across many scientific disciplines, notably through quantum neural network (QNN) [Schuld et al.(2014),Biamonte et al.(2017)]. Parameterized quantum circuits (PQCs) have emerged as a central component of QNNs [Benedetti et al.(2019)], which are often trained similarly to classical deep neural networks (DNNs) to approximate functions that map input features to output values. Recent studies have exploited PQCs to tasks traditionally performed by classical DNNs [Hur et al.(2022),Bausch(2020),Chen et al.(2020)]. While this line of research is valuable, it is equally important to uncover the *intrinsic characteristics* of PQCs. As argued by [Schuld and Killoran(2022)], with the discovery of the unique capabilities of PQCs, we can move towards realizing a *genuine quantum supremacy* for QNN by applying PQCs to tasks that are naturally suited to their quantum nature.

**Related Work.** PQCs typically consist of a sequence of quantum gates, which are elementary unitary operations acting on quantum states. Despite recent advancements in QNN, the design principle of PQC remains more of an art than a mathematically grounded discipline, as the sequence of quantum gates are often chosen arbitrarily. Recent research has begun addressing this by developing theoretical foundations for principled design [Nguyen et al.(2024),Yu et al.(2024)]. For example, [Pérez-Salinas et al.(2020)] introduced a data reuploading design and rigorously establish an analogy between classical DNNs and PQCs. Expanding on this, [Schuld et al.(2021)] showed that PQCs with the data reuploading structure enables the resulting quantum model's output to be represented as partial Fourier series. More recently, [Yu et al.(2022)] proved the universal expressivity of single-qubit QNN for approximating bounded univariate functions. However, there remains a fundamental gap in understanding the variational power of PQCs, especially when the target function is purely a black-box input. [Yu et al.(2022)] assumed prior knowledge of the target function's Fourier coefficients when constructing PQC to demonstrate its universal approximation capability. However, this limits the model's applicability in practical learning scenarios. As a result, a key open question is to understand how PQCs learn directly through training, without any prior knowledge of the target function's Fourier coefficients.

**Overview.** This paper presents both theoretical and empirical analysis to explain how the design of PQCs influence the Fourier coefficients of the quantum model's output. By viewing the state evolution of a qubit under a PQC as a trajectory on the Bloch sphere, we establish a formal mathematical connection between this trajectory and the Fourier coefficients of the target function. Specifically, we show that different PQC configurations shape distinct qubit trajectories, which directly determine the resulting Fourier coefficients. Leveraging this interpretation, we propose insights to systematically design PQC architectures for enhancing the model's learning ability. Furthermore, to extend this

understanding into actionable design, we further introduce a principled method for constructing sophisticated data-encoding Hamiltonian with tunable scalers. This allows us to select spectra aligned with desired frequencies and enable a more deliberate control over the frequency content of the model's output.

**Contributions.** The main contributions of this research are two-fold as follows. First, we provide empirical and mathematical evidence that a single-layer PQC learns a target function by modulating the radius of a circular trajectory traced by the qubit on the Bloch sphere. Furthermore, using a Lie algebraic approach [Kharchenko(2015)], we establish a theorem that formally connects the trajectory radius, the corresponding Fourier coefficient, and the measurement axis.

## 2    Quantum Model and Preliminaries

**Qubit.** A quantum bit, short for qubit, is a fundamental unit of quantum information. Mathematically, a qubit is described by a unit vector in a two-dimensional Hilbert space, written as $|\psi\rangle = \alpha |0\rangle + \beta |1\rangle$, where $|0\rangle = [1, 0]^\top$ and $|1\rangle = [0, 1]^\top$ are the computational basis states and $\alpha, \beta \in \mathbb{C}$ are complex-valued probability amplitudes that satisfy $|\alpha|^2 + |\beta|^2 = 1$. A qubit state is visualized in the *Bloch sphere*, where a single-qubit is represented as a point on the surface of a unit sphere in three-dimensional space. Therefore, any qubit state can be parameterized as $|\psi\rangle = \cos(\theta/2) |0\rangle + e^{i\phi} \sin(\theta/2) |1\rangle$, where $\theta \in [0, \pi]$ and $\phi \in [0, 2\pi)$.

**Parameterized Quantum Circuit.** QNNs are quantum analog of classical neural networks, designed to leverage quantum principles such as superposition to learn patterns and representations from data [Gawron and Lewiński(2020)]. A widely adopted model for constructing QNNs is the PQC. PQC is composed of sequence of quantum gates whose rotation angles depend on classical input data or tunable parameters that are optimized during training. Mathematically, PQC implements unitary transformation $U(x, \boldsymbol{\theta})$ acting on an initial state $|0\rangle^{\otimes n}$, where $x \in \mathbb{R}^d$, $\boldsymbol{\theta} = (\theta_1, \ldots, \theta_p) \in \mathbb{R}^p$ and $n$ is the input data, the trainable parameters, and number of qubits, respectively. We consider a PQC architecture consisting of alternating sequences of data-encoding gates $S(x)$ and parameterized trainable block $W(\boldsymbol{\theta})$ [Pérez-Salinas et al.(2020)], such that the total unitary operator is described as, $U(x, \boldsymbol{\theta}) = W^{L+1}(\boldsymbol{\theta})S(x)W^L(\boldsymbol{\theta}) \ldots S(x)W^1(\boldsymbol{\theta})$, where $L$ denotes the number of reuploading gates. Each trainable block $W^l$ typically consists of cascaded parameterized Pauli rotation gates, i.e., $R_X(\theta) = e^{-i\frac{\theta}{2}\sigma_X}$, $R_Y(\theta) = e^{-i\frac{\theta}{2}\sigma_Y}$, and $R_Z(\theta) = e^{-i\frac{\theta}{2}\sigma_Z}$, where $\sigma_X = \begin{bmatrix} 0 & 1 \\ 1 & 0 \end{bmatrix}$, $\sigma_Y = \begin{bmatrix} 0 & -i \\ i & 0 \end{bmatrix}$, and $\sigma_Z = \begin{bmatrix} 1 & 0 \\ 0 & -1 \end{bmatrix}$ respectively, are Pauli-$X$, Pauli-$Y$, and Pauli-$Z$ operators.

**Measurement and Projection.** When a qubit is measured in the computational basis, it collapses to either $|0\rangle$ or $|1\rangle$ with probabilities $|\alpha|^2$ and $|\beta|^2$. This process is probabilistic and irreversibly alters the superposition state of the qubit. The measurement takes place via an observable, typically selected as $\sigma_Z$, whose eigenvectors are $|0\rangle$ and $|1\rangle$ with corresponding eigenvalues of $\pm 1$. The output of

a quantum model is the expectation value of this observable $M$ with respect to the quantum state processed by a PQC, i.e., $f_{\boldsymbol{\theta}}(x) = \langle 0 | U^{\dagger}(x, \boldsymbol{\theta}) M U(x, \boldsymbol{\theta}) | 0 \rangle$. For the observable $\sigma_Z$, this expectation value evaluates to $f_{\boldsymbol{\theta}}(x) = |\alpha|^2 - |\beta|^2$. Geometrically, this expectation value corresponds to the projection of the qubit's state onto the measurement axis $\hat{z}$. Hence, observable $\sigma_Z$ confines the model's output within the range $[-1, 1]$, defining a bounded, real-valued function: $f_{\boldsymbol{\theta}}(x) : \mathbb{R}^d \to [-1, 1]$.

**Partial Fourier Series Analysis.** Consider a unitary operator that adopts a data reuploading structure [Pérez-Salinas et al.(2020)]. Each data-encoding step is performed by a Pauli gate-based encoder, given by $S(x) = e^{-ix\boldsymbol{H}}$, where $\boldsymbol{H}$ is the Hamiltonian that governs the encoded data's evolution. The spectral decomposition of this Hamiltonian determines the accessible frequency spectrum of the quantum model's output. As shown by [Schuld et al.(2021)], the resulting output of such PQCs naturally can be expressed as partial Fourier series described as $f(x) = \sum_{\omega \in \Omega} c_{\omega} e^{i\omega x}$, where the set of accessible frequencies $\Omega$ arises from differences between eigenvalues of the Hamiltonian and the Fourier coefficients $c_{\omega}$ depends on the circuit parameters and measurement choices. [Yu et al.(2022)] proved that a single qubit quantum circuit can act as a universal approximator for bounded univariate functions. However, they are limited for providing intuitive or structural explanations of how individual PQCs affect the resulting Fourier coefficients.

# 3   Designing the PQC

## 3.1   Projected Circle Theorem

To gain a more general understanding, we now present a theorem that formalizes this learning behavior. This offers guidance for designing PQC to approximate arbitrary target functions. As first step, we base our analysis towards three-dimensional geometric perspective, since every qubit state can be represented as a point on the Bloch sphere. In particular, we invoke Rodrigues' formula [[Dai(2015)]].

**Lemma 1.** *(**Rodrigues' Formula**) Let $\mathbf{n} \in \mathbb{R}^3$ be a unit axis, i.e., $||\mathbf{n}|| = 1$, and let $R_{\mathbf{n}}(\alpha) \in SO(3)$ be the rotation by angle $\alpha$ around the axis $\mathbf{n}$. Then for any vector $\mathbf{u} \in \mathbb{R}^3$, $R_{\mathbf{n}}(\alpha)\mathbf{u} = \mathbf{u}\cos\alpha + (\mathbf{n} \times \mathbf{u})\sin\alpha + (\mathbf{n} \cdot \mathbf{u})\mathbf{n}(1 - \cos\alpha)$.*

*Proof.* Consider an arbitrary rotation about axis $w = [w_x, w_y, w_z]^{\top}$ when $\|w\|^2 = 1$. Then, set of all possible rotations satisfying aforementioned condition is $SO(3)$ group. Consider a Lie algebra of $SO(3)$, $\mathfrak{so}(3)$. $\hat{w} \in \mathfrak{so}(3)$, which is a corresponding element of $w$, can be expressed as follows,

$$\hat{w} = \begin{bmatrix} 0 & -w_z & w_x \\ -w_y & 0 & w_y \\ -w_x & w_z & 0 \end{bmatrix}. \tag{1}$$

Rotating a vector $u \in \mathbb{R}^3$ about $\theta$ around the axis of $w$ can be expressed as $\theta \hat{w}$. Corresponding rotation $R \in SO(3)$ of $\theta \hat{w} \in \mathfrak{so}(3)$ is defined as follows,

$$\exp(\theta \hat{w}) = \mathbf{I} + \theta \hat{w} + \frac{1}{2!}\theta^2 \hat{w}^2 + \frac{1}{3!}\theta^3 \hat{w}^3 + \cdots . \tag{2}$$

Using the identity $\hat{w}^2 = -\mathbf{I} + \hat{w}\hat{w}^\mathsf{T}$,

$$\exp(\theta \hat{w}) = \mathbf{I} + (1 - \frac{\theta^3}{3!} + \frac{\theta^5}{5!} + \cdots)\hat{w} + (-\frac{\theta^2}{2!} + \frac{\theta^4}{4!} - \frac{\theta^6}{6!} + \cdots)\hat{w}^2 \tag{3}$$

$$= \mathbf{I} + \sin\theta \hat{w} + (1 - \cos\theta)\hat{w}^2. \tag{4}$$

When applied $R = \exp(\theta \hat{w})$ on vector $u$, rotated vector $u_{\mathbf{rot}}$ is defined as,

$$u_{\mathbf{rot}} = Ru = u + \sin\theta \hat{w}u + (1 - \cos\theta)\hat{w}^2 u. \tag{5}$$

Note that $\hat{w}u = w \times u$. Also, by formula $a \times (b \times c) = b(a \cdot c) - c(a \cdot b)$,

$$\hat{w}^2 u = \hat{w}(w \times u) = w \times (w \times u) = w(w \cdot u) - u(w \cdot w) = w(w \cdot u) - u. \tag{6}$$

Replacing $\hat{w}u$ and $\hat{w}^2 u$ in (5) using (6) completes the proof.

In general, a rotation about an arbitrary unit axis vector $\mathbf{n} \in \mathbb{R}^3$ by an angle $x$ is represented by the rotation operator $R_\mathbf{n}(\alpha)$. Specifically, the action of $R_\mathbf{n}(\alpha)$ on a vector $\mathbf{u}$ describes rotating $\mathbf{u}$ around the axis $\mathbf{n}$. However, Lemma 1 only applies to rotations acting on vectors in $\mathbb{R}^3$. To justify its relevance in our setting, we note that every Pauli-gate belongs to the $SU(2)$ group, which is a set of all the spatial rotations of a vector on the Bloch sphere [Hall(2013)]. Since this vector evolves within $\mathbb{R}^3$, the qubit state evolution under a single qubit unitary can be entirely captured using $SO(3)$ rotations. The following lemma formalizes this connection.

**Lemma 2. (*Adjoint Homomorphism*)** *Assume a mapping $f : \mathbb{C}^{2 \times 2} \to \mathbb{R}^3$ such that, $f : \rho \mapsto \mathbf{tr}(\rho \sigma_X), \mathbf{tr}(\rho \sigma_Y, \mathbf{tr}(\rho \sigma_Z))$, where $\mathbf{tr}(\cdot)$ denotes the trace operator. Then, for $\forall R \in SU(2)$, $\exists R' \in SO(3)$ such that $f(R\rho R^\dagger) = R'f(\rho)$.*

*Proof.* Consider an arbitrary density matrix $\rho$. Then, it can be analyzed as follows, $\rho = \frac{1}{2}(\mathbf{I} + r \cdot \sigma)$, where $r = [r_1, r_2, r_3]^\top$ and $r \cdot \sigma = r_1 \sigma_X + r_2 \sigma_Y + r_3 \sigma_Z$. Note that $\mathbf{tr}(\sigma_a \sigma_b) = \delta_{ab}$ for $a, b \in \{x, y, z\}$ and $\delta_{ab}$ denotes Kronecker delta function. Then,

$$f(\rho) = (\mathbf{tr}(\rho \sigma_X), \mathbf{tr}(\rho \sigma_Y), \mathbf{tr}(\rho \sigma_Z)) = (r_1, r_2, r_3) = r. \tag{7}$$

Thus function $f$ maps arbitrary density matrix to a vector on three-dimensional ball. Consider an arbitrary rotation $R \in SU(2)$ acts on the density matrix. Then,

$$R\rho R^\dagger = \frac{1}{2}(\mathbf{I} + R(r \cdot \sigma)R^\dagger). \tag{8}$$

Here, note that $r \cdot \sigma$ is also an element of $\mathfrak{su}(2)$ group, since every density matrix can be expressed as linear combination of Pauli matrices and Identity matrix.

Thus, $R(r \cdot \sigma)R^{\dagger}$ in above equation can be interpreted as an adjoint representation of $SU(2)$. Due to the property of $\mathfrak{su}(2)$, $\forall R\sigma_i R^{\dagger} \in \mathfrak{su}(2)$, there exist rotation in $r' \in \mathbb{R}^3$ such that,

$$R\sigma_i R^{\dagger} = r' \cdot \sigma. \tag{9}$$

Thus, below equation holds,

$$f(R\rho R^{\dagger}) = f(\frac{1}{2}(\mathbf{I} + R(r \cdot \sigma)R^{\dagger})) \tag{10}$$

$$= f(\frac{1}{2}\mathbf{I} + r_1 R\sigma_X R^{\dagger} + r_2 R\sigma_Y R^{\dagger} + r_3 R\sigma_Z R^{\dagger}) \tag{11}$$

$$= f(\frac{1}{2}\mathbf{I} + r_1(r_1' \cdot \sigma_X) + r_2(r_2' \cdot \sigma_Y) + r_3(r_3' \cdot \sigma_Z)) \tag{12}$$

$$= f(\frac{1}{2}\mathbf{I} + (R'r) \cdot \sigma) = R'f(\rho), \tag{13}$$

where $R'$ is $3 \times 3$ matrix whose column vectors are $r_1', r_2', r_3'$. This completes the proof.

Consider a single-qubit unitary operator of $U(x, \boldsymbol{\theta}, \boldsymbol{\phi}) = V_L(\boldsymbol{\theta})R_X(x)V_R(\boldsymbol{\phi})$, where $R_X(x)$ is the Pauli-$X$ gate. The operators $V_L(\boldsymbol{\theta})$ and $V_R(\boldsymbol{\phi})$ are products of Pauli gates, $R_Q(\eta) = \exp(-i\eta Q/2)$, where $Q = \{\sigma_X, \sigma_Y, \sigma_Z\}$, with trainable parameters collected in vectors $\boldsymbol{\theta} \in \mathbb{R}^N$ and $\boldsymbol{\phi} \in \mathbb{R}^M$. Here, $N$ and $M$, respectively, are the numbers of trainable parameters in $V_L$ and $V_R$. By Lemma 2, $V_L, V_R \in SU(2)$ induce transformation in $\mathbb{R}^3$ as $R_{V_L}, R_{V_R} \in SO(3)$, respectively.

**Theorem 1.** *(**Projected Circle**) Let $\mathbf{n} \in \mathbb{R}^3$ be a unit vector (the data-encoding axis), $\hat{z} = [0, 0, 1]^{\top}$ be the initial point of qubit state, and $R_{V_R}\hat{z} = \mathbf{u} \in \mathbb{R}^3$ be a rotated initial vector for qubit in $\mathbb{R}^3$. By Lemma 1, consider the rotation of $\mathbf{u}$ by angle $x$ about $\mathbf{n}$ as $\boldsymbol{r}(x) = R_{\mathbf{n}}(x)\mathbf{u}$. Let $R_{V_L} \in SO(3)$ be another rotation. Then, the output of the quantum model is described as,*

$$f(x) = \hat{z} \cdot (\mathcal{R}_{V_L}R_{\mathbf{n}}(x)\mathbf{u}) = c_z + v_z \cos x + w_z \sin x, \tag{14}$$

*where the model spans $\{1, \cos x, \sin x\}$ via coefficients $c_z, v_z, w_z \in \mathbb{R}$ that depend on $\mathbf{u}, \mathbf{n}$ and $R_{V_L}$. The function is constant if $v_z = w_z = 0$; otherwise, both $\cos x$ and $\sin x$ appear.*

*Proof.* By Lemma 1, as $x$ varies, $\mathbf{u}$ rotates around $\mathbf{n}$ as

$$\boldsymbol{r}(x) = R_{\mathbf{n}}(x)\mathbf{u} = \mathbf{u}\cos x + (\mathbf{n} \times \mathbf{u})\sin x + (\mathbf{n} \cdot \mathbf{u})\mathbf{n}(1 - \cos x), \tag{15}$$

tracing out a circle in $\mathbb{R}^3$. Applying the fixed rotation $\mathcal{R}_{V_L}$ to $\boldsymbol{r}(x)$ and projecting it onto $\hat{z}$ yields $f(x) = \underbrace{\hat{z} \cdot \mathcal{R}_{V_L}\mathbf{u}}_{c_z}\cos x + \underbrace{\hat{z} \cdot \mathcal{R}_{V_L}(\mathbf{n} \times \mathbf{u})}_{w_z}\sin x + \underbrace{\hat{z} \cdot \mathcal{R}_{V_L}\mathbf{n}(\mathbf{n} \cdot \mathbf{u})}_{d_z}(1 - \cos x)$. If $v_z \triangleq d_z(\mathbf{n} \cdot \mathbf{u}) - c_z$, $f(x)$ matches (14). Then, coefficients of $\cos x$ and $\sin x$ vanish if $\hat{z}$ is orthogonal to both $\mathcal{R}_{V_L}\mathbf{u}$ and $\mathcal{R}_{V_L}(\mathbf{n} \times \mathbf{u})$. This completes the proof.

**Corollary 1.** *The radius of the circle is determined by $\|\mathbf{u} \times \mathbf{n}\|$.*

*Proof.* The unit vector $\mathbf{u}$ can be decomposed to $\mathbf{u} = \mathbf{u}_{\parallel} + \mathbf{u}_{\perp}$, where $\mathbf{u}_{\parallel}$ and $\mathbf{u}_{\perp}$ are parallel and orthogonal components of $\mathbf{u}$. If $\mathbf{u} \parallel \mathbf{n}$, then $\mathbf{u} = \mathbf{u}_{\parallel}$. Now, $\mathbf{u}_{\parallel}$ is projection of $\mathbf{u}$ onto $\mathbf{n}$. Then, $\mathbf{u} = (\mathbf{n} \cdot \mathbf{u}) \frac{\mathbf{n}}{||\mathbf{n}||} = ||\mathbf{u}|| \, \mathbf{n}$. Further, $(\mathbf{n} \cdot \mathbf{u})\mathbf{n} = ||\mathbf{u}|| \, \mathbf{n}$ since $\mathbf{n}$ is a unit vector. We apply these in (15) and then,

$$\boldsymbol{r}(x) = ||\mathbf{u}||\mathbf{n}(\cos x + 1 - \cos x) = ||\mathbf{u}||\mathbf{n}. \tag{16}$$

Therefore, (15) collapses to $||\mathbf{u}||\mathbf{n}$, which is a single vector independent to $x$. This completes the proof.

From Corollary 1, we note that the radius is fully determined by the vector $\mathbf{u}$, which makes it our primary design choice. Therefore, to effectively control the radius of the trajectory, it is essential to have flexible and unrestricted control over the selection of $\mathbf{u}$.

**Corollary 2.** *If $\mathbf{u}$ and $\mathbf{n}$ are parallel, i.e., $\mathbf{u} \parallel \mathbf{n}$, $\boldsymbol{r}(x)$ degenerates to a single point.*

*Proof.* Decompose $\mathbf{u}$ into its parallel and orthogonal components, $\mathbf{u} = \mathbf{u}_{\parallel} + \mathbf{u}_{\perp} = (\mathbf{n} \cdot \mathbf{u})\mathbf{n} + [\mathbf{u} - (\mathbf{n} \cdot \mathbf{u})\mathbf{n}]$. Because the rotation axis is $\mathbf{n}$, the orthogonal component $\mathbf{u}_{\perp}$ is exactly what sweeps out a circle when $\mathbf{u}$ is rotated. Equivalently, $\mathbf{u} \times \mathbf{n}$ is a vector orthogonal to $\mathbf{n}$ whose magnitude is equal to magnitude of $\mathbf{u}_{\perp}$ as $||\mathbf{u} \times \mathbf{n}|| = ||\mathbf{u}|| \, ||\mathbf{n}|| \sin(\pi/2) = ||\mathbf{u}_{\perp}||$. Therefore, the circle is traced out by $\mathbf{u}_{\perp} \cos x + (\mathbf{n} \times \mathbf{u}_{\perp}) \sin x$ whose magnitude is exactly $||\mathbf{u} \times \mathbf{n}||$. This completes the proof.

**Corollary 3.** *(**L-Layer PQC**) Consider $L$ consecutive sequence of PQC layers: $U(x, \boldsymbol{\theta}) = W^{L+1}(\boldsymbol{\theta})S(x)W^{L}(\boldsymbol{\theta}) \ldots S(x)W^{1}(\boldsymbol{\theta})$, which is equivalent to sequence of single-qubit rotations $\prod_{i=1}^{L} \mathcal{R}_{V_L}^{i} R_{\mathbf{n}_i}(x) \mathcal{R}_{V_R}^{i}$, each by same angle $x$ about some different unit axis $\mathbf{n}_l \in \mathbb{R}^3$. Let $\boldsymbol{r}_l(x) = R_{\mathbf{n}_l}(x)[\boldsymbol{r}_{l-1}(x)]$ and $\boldsymbol{r}_0(x) = \mathbf{n}$. Then, $\boldsymbol{r}_L(x) \in span\{1, \sin x, \cos x, \ldots, \sin(Lx), \cos(Lx)\}$.*

*Proof.* We prove Corollary 3 by mathematical induction on layer $l$.
   *Base case:* When $l = 1$, the analysis is equivalent to Theorem 1 itself.
   *Inductive step:* Assume for $l - 1$ layers we already have

$$\boldsymbol{r}_{l-1}(x) \in \text{span}\{1, \sin x, \cos x, \sin(2x), \cos(2x), \ldots, \sin((l-1)x), \cos((l-1)x)\}. \tag{17}$$

Now we apply the next rotation for $l$-th rotation: $\boldsymbol{r}_l(x) = R_{\mathbf{n}_l}(x)\boldsymbol{r}_{l-1}(x)$. This, by Rodrigues' formula is just another rotation of $\boldsymbol{r}_{l-1}$ with respect to data-encoder rotation axis $\mathbf{n}$. Thus, $\boldsymbol{r}_l(x) = \boldsymbol{r}_{l-1}(x)\cos x + [\mathbf{n} \times \boldsymbol{r}_{l-1}(x)]\sin x + [\mathbf{n} \cdot \boldsymbol{r}_{l-1}(x)]\mathbf{n}(1 - \cos x)$. Since each term $\boldsymbol{r}_{l-1}(x)$ is a sum of harmonics up to $(l-1)$ in $\sin x, \cos x$, multiplying one more by $\{\sin x, \cos x, (1 - \cos x)\}$ can produce up to $l$-th order harmonic terms. Specifically,

$$\sin((l-1)x)\cos x = 1/2\left[\sin(lx) + \sin((l-2)x)\right], \tag{18}$$

$$\cos((l-1)x)\cos x = 1/2\left[\cos(lx) + \cos((l-2)x)\right], \tag{19}$$

$$\sin((l-1)x)\sin x = 1/2\left[\cos((l-2)x) - \cos(lx)\right]. \tag{20}$$

Thus, $\boldsymbol{r}_l(x) \in \text{span}\{1, \sin(kx), \cos(kx) | k \leq l\}$. This completes the inductive proof.

From Corollary 3, we remark that the final qubit trajectory induces complicated multi-loop trajectory because each layer pushes the rotation to different axis while sharing the same $x$.

**PQC Designing Insight.** Building on the Theorem 1, we highlight key design insights. **i)** *Choose a non-collinear* **u** *to* **n***:* To avoid Corollary 2, **u** must be designed such that it *must not be* collinear with the data-encoding axis **n**. **ii)** *Avoid Degenerate Projections:* As the *remark* notes, if the final rotation leaves the Bloch vector primarily parallel to the $xz$-plane, then projecting onto the $yz$-plane degenerates the circle to a line, which loses the trajectory's expressivity. These insights emphasize *flexible* design of rotation axes, which allows robust tilting of the rotating vector the circuit's expressiveness is well preserved. This intuition extends naturally to $L$-layer circuits: introducing flexible and independent rotation axes at each layer enhances the model's ability to produce richer trajectories, enabling effective representation of higher harmonics and more complex target functions.

## 4   Conclusion

In this paper, we investigated the intrinsic structure and behavior of PQCs from the perspective of output qubit trajectories. Through theoretical and empirical analyses, we demonstrated that the configuration of trainable parameters directly govern the radius and orientation of the output trajectory on the Bloch sphere. We formally established a rigorous mathematical connection between the qubit trajectory, measurement axis, and the resulting Fourier coefficients. These results significantly bridge the gap between quantum circuit structure, geometric insights from trajectory analysis, and the practical design considerations necessary for PQC's full expressive potential.

## 5   Acknowledgments

The complete version of this paper is currently under review at the Conference on Neural Information Processing Systems (NeurIPS), San Diego, USA, December 2025. The corresponding authors of this paper are Prof. Soohyun Park and Prof. Joongheon Kim (e-mail: soohyun.park@sookmyung.ac.kr, joongheon@korea.ac.kr).

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
