# OpenReview forum: "Understanding the Qubit Trajectory in Parameterized Quantum Circuits"
_purdue.edu/Purdue_University/PQAI/2025/Symposium — PQAI 2025 Oral_

### Official Review · Reviewer_m69u · 2025-07-23
**A solid and well-motivated analysis of single-qubit PQCs with useful design insights—acceptable, though it lacks discussion on scalability and hardware noise.**

**Rating:** 7
**Confidence:** 4

**Review:**

This paper looks at how parameterized quantum circuits (PQCs) work by studying how the trainable parameters shape the qubit’s path on the Bloch sphere. It connects this trajectory to Fourier coefficients using Lie algebra and shows how circuit design choices—like rotation axes—affect expressivity.

The strengths are its clear geometric perspective, solid math, and practical design tips. But it mainly focuses on single-qubit cases and doesn’t explore noise or real-device training.

---

### Decision · Program_Chairs · 2025-07-29

Accept (Oral)